# Formulation and validation of a regional household wealth index for sub-Saharan Africa

**Moaven Razavi**[ID]*, **Collins Gaba, William Crown**[ID], **Allyala Nandakumar**

Institute for Global Health and Development, The Heller School for Social Policy and Management, Brandeis University, Waltham, Massachusetts, United States of America

* mrazavi@brandeis.edu

## Abstract

A new era in global health assistance requires a focus on efficiently using limited and declining donor funds. This shift requires better evaluation methods to allocate resources effectively. Most evaluations in low- and middle-income countries (LMICs) examine health disparities within countries, but it is also crucial to assess health outcomes at an inter-country level based on national wealth. Cross-country studies support resource reallocation to the neediest nations and help transition programs like HIV responses within countries with better health infrastructure. This paper presents an unsupervised machine learning method, Principal Component Analysis (PCA), applied to household surveys from 15 African countries to create a universal wealth index that allows multiple countries to be compared on a common scale. Our method places households on a regional wealth scale, enabling cross-country comparisons of health indicators. We used a pooled dataset of 136,086 households from 15-Population-based HIV Impact Assessment (PHIA) countries and validated our universal ranking approach against a local wealth indicator adjusted for macroeconomic differences. The results showed coherence between the macroeconomic-adjusted multinational scale and the PCA-created regional scale, supporting the method's usability for regional household rankings. The proposed method relocates households, as citizens of the world, on a regional wealth scale compared to most surveys that rank them by income placements in their local states. The validation results suggest that the direction and magnitude of mobility of households from national to regional scale in both methods were adequately coherent, ensuring the usability of our approach in ranking households regionally. The PCA-created border-agnostic wealth quintiles enable policymakers to optimize their efficiency improvement efforts, which promises superior efficiency gains over the siloed localized efficiency improvements. Our approach, tested on PHIA-participating countries, can be replicated for similar surveys to study utilization patterns and health outcomes globally.

**Data availability statement:** The data that support the findings of this study are available from PHIA project at Columbia University. According to our university sponsored data use agreement we may not publicly share the data on behalf of the PHIA project. However, researchers can create an account and login information within the PHIA PROJECT website and access the country/year specific data for individual countries. The URL of the data repository is https://phia-data.icap.columbia.edu/datasets.

**Funding:** The contents of this manuscript were developed under grant INV-046280 from the Bill & Melinda Gates Foundations. The funders had no role in study design, data collection and analysis, decision to publish, or preparation of the manuscript. The content is solely the responsibility of the authors and does not necessarily represent the official views of the funding foundation.

**Competing interests:** The authors have declared that no competing interests exist.

## Introduction

As the transition of HIV care to country ownership progresses, it is important to ensure data-driven policies guide this shift to preserve past achievements [1,2]. Since its founding in 2003, the PEPFAR program has saved over 25 million lives and granted over 20.5 million people access to antiretroviral therapy [3]. To sustain these gains as resources become constrained, several intra-country analyses have been conducted to guide the reallocation of funds and resources within comparative groups with poorer access to HIV treatment or outcomes. Researchers have used this approach to address access and outcome gaps in HIV treatment based on urban/rural residence, wealth, sex, and age [4]. Although such intra-country studies are useful, an approach is needed that facilitates reallocation analyses across multiple countries in a consistent fashion.

Studies indicate that wealthier individuals tend to have better outcomes within countries [5–7]. Asset-based wealth indices are commonly used to assess households' socioeconomic status in low- and middle-income countries (LMICs) when reliable income data is unavailable. Different approaches exist to measure wealth and poverty, including relative measures such as the DHS wealth quintiles [8,9] and absolute measures such as the World Bank's international poverty thresholds [10,11]. Relative measures are useful for assessing inequality within countries [12], while absolute measures enable cross-country comparisons but may overlook within-country disparities [13]. Our study builds on the relative approach by creating a regional index that allows households across sub-Saharan Africa to be ranked on the same scale. Surveys like the Demographic Health Survey (DHS) provide accessible data, making wealth measurements in LMICs reliable. These methods effectively explain variations in household health status and outcomes [14–16]. The link between wealth and HIV services has prompted discussions on reallocating resources to underserved populations [4,17–19]. Much of this research has been done using country-level household wealth indices [20,21].

We plan to create a universal wealth proxy for intercountry resource reallocation, allowing one to rank households in the 15 Population-based HIV Impact Assessment (PHIA) countries included in this study on a regional scale. This will enable us to investigate wealth-health correlations consistently across regions. By regionally positioning households, one can examine HIV access and outcomes distribution without borders. Our goal is to achieve equality in HIV access and outcomes worldwide and identify inequity patterns to inform policies and resource reallocation with less funding for HIV programs.

We construct the universal household wealth index using Principal Component Analysis (PCA), which creates a wealth proxy using asset ownership data from the pooled PHIA survey dataset. The first principal component's asset weights are pooled and ranked to create the universal index [15,20].

The universal index is generated based on pooled data collected between 2016 and 2021 from the 15 PHIA participating countries in Africa, totaling 136,086 households. This text will also discuss the methodology used in creating this index, as well as the approach to testing and validation of the regional index to ensure its stability and applicability for research.

## Methodology

### Data

We pooled data from the PHIA Surveys conducted by the US President's Emergency Plan for AIDS Relief (PEP-FAR) through the Centers for Disease Control and Prevention (CDC). The data came from 15 PHIA participating countries: Botswana (2021), Cameroon (2017), Cote D'Ivoire (2017), Eswatini (2020), Ethiopia (2017), Kenya (2018), Lesotho (2020), Malawi (2020), Mozambique (2021), Namibia (2017), Rwanda (2018), Tanzania (2016), Uganda (2020), Zambia (2021), Zimbabwe (2020). Data from all countries were pooled without applying population-size weights, following approaches used in prior cross-country wealth index research [15], which has shown that PCA-based wealth indices are highly robust to whether countries are weighted equally or by population size. While two waves of PHIA surveys were available for seven of those countries, data from only six was available for analysis.

Informed by the literature, we analyzed 24 household survey variables to predict wealth, including assets such as mobile phones, radios, televisions, refrigerators, cars, livestock, housing materials, electricity, clean water, sanitation, cooking fuel, rooms per capita, and urban interaction term. We also considered individual factors like the head of the household's education and their relationship with other household members.

This study's pooled dataset contains 136,086 households from 15 PHIA participating countries. As part of our universal wealth index validation efforts, data on gross domestic product (GDP) per capita [purchasing power parity (PPP), current international$] were collected for the 15 countries from the World Development Indicators [22] (WDI) database. Table 1 lists the 15 countries with their macroeconomic indicator.

**Table 1. PHIA countries with their macroeconomic indicator.**

| Country (PHIA round and year) | PHIA Survey Sample size | 2019 (GDP per capita, PPP (current international$) | Macro-Economic Adjuster |
|---|---|---|---|
| **Botswana (PHIA1–2021)** | 7,011 | 14,890 | 5.05 |
| **Cameroon (PHIA1–2017)** | 9,635 | 4,195.67 | 1.42 |
| **Cote d'Ivoire (PHIA1–2017)** | 7,349 | 5,948 | 2.02 |
| **Eswatini (PHIA2–2020)** | 5,036 | 9,083 | 3.08 |
| **Ethiopia (PHIA1–2017)** | 9,422 | 2,274 | 0.77 |
| **Kenya (PHIA1–2018)** | 11,920 | 4,711 | 1.60 |
| **Lesotho (PHIA2–2020)** | 7,911 | 2,550 | 0.87 |
| **Malawi (PHIA2–2020)** | 12,235 | 1,466 | 0.50 |
| **Mozambique (PHIA1–2021)** | 7,498 | 1,370 | 0.47 |
| **Namibia (PHIA1–2017)** | 7,234 | 10,410 | 3.53 |
| **Rwanda (PHIA1–2018)** | 9,604 | 2,325 | 0.79 |
| **Tanzania (PHIA1–2016)** | 12,993 | 2,947 | 1.00 |
| **Uganda (PHIA2–2020)** | 9,319 | 2,444 | 0.83 |
| **Zambia (PHIA2–2021)** | 9,054 | 3,386 | 1.15 |
| **Zimbabwe (PHIA2–2020)** | 9,865 | 2,712 | 0.92 |

Notes:

$Macroeconomic\ Adjuster = \frac{Country\ GDP\ per\ capita,\ PPP(current\ international\ \$)}{Median\ Country\ GDP\ per\ capita,\ PPP(current\ international\ \$)}$

Median Country = Tanzania.

GDP = Gross Domestic Product; PHIA = Population-based HIV Impact Assessment; PPP = Purchasing Power Parity.

PHIA1 = first wave of PHIA survey; PHIA2 = second wave of PHIA survey.

## Inclusion criteria

We excluded households with missing values in key variables, as well as responses marked "Don't Know" (−8) or "Refused" (−9) for asset ownership and socioeconomic data. We analyzed data collected only from the head of the household.

## Variable transformations

Several transformations were applied to ensure data consistency and comparability:

- Binary variables: asset ownership variables (e.g., mobile phone, radio, television, refrigerator, electricity, car, livestock ownership) were recoded into 0 = No, 1 = Yes.

- Ordinal variables: categorical variables such as water source, toilet type, roofing material, flooring material, and cooking fuel were ranked hierarchically, with higher values representing better living conditions.

- Education variable: the categorical education variable was transformed into years of schooling. For example, the transformation used for Mozambique was as follows: No formal education = 0 years, Primary education = 6 years, Secondary education = 12 years, Tertiary education = 16 years. This was slightly modified based on each country's educational system.

- New variables created:

  ◦ Room per capita: defined as the number of sleeping rooms divided by the number of household members.

  ◦ Urban room per capita: interaction term between room per capita and urban residency to differentiate the valuation of housing in urban areas.

## Principal Component Analysis (PCA) for creation of wealth index

PCA is a popular unsupervised machine learning method for ranking objects in a multi-dimensional space based on their characteristics. It generates orthogonal eigenvectors, each representing a latent variable of the population. According to empirical literature, the first eigenvector PC1, which corresponds to the largest eigenvalue of the covariance matrix of household variables, most effectively differentiates households by their wealth and economic wellbeing. PC1, the first eigenvector, is considered a proxy for the unmeasured latent variable wealth in the PHIA survey as it weights and combines all household characteristics in a manner that the resulting score for each household represents its relative ranking in a wealth scale. To generate such wealth index, we conducted PCA on the set of asset ownership and socioeconomic variables explained earlier and calculated the PC1 eigenvector for each household. The wealth scores generated by PC1 were used to rank and classify households into five equally sized wealth quintiles (Q1–Q5). The PCA method was implemented for each country and then for the pooled sample of all countries separately. Separately, the PHIA survey provides a wealth indicator and a wealth quintile for each household. To assess the consistency of the PC1 scores, the correlation between our PCA-derived wealth index and the original PHIA wealth quintile was calculated. The Pearson Correlation between the two ranking schedules ranged from 78% for Eswatini to 94% for Lesotho, with an average of 89%. This indicates that our generated PC1 scores for each country exhibit a strong correlation with PHIA's wealth scores. Our final PCA model on the combined dataset produced PC1 scores for all 136,086 households across the 15 participating PHIA countries, enabling us to rank and classify households into five equally sized wealth quintiles (Q1–Q5).

## Test of the stability of the wealth index's underlying components

We examined the stability of the components shaping the PC1 scores. The PC1 vector is a linear combination of household characteristics, each contributing uniquely. Some characteristics positively affect the PC1 while others negatively

impact it, and are determined by weights assigned to each household characteristic. To ensure consistency across countries, a stability test verified that household characteristics like urban house size or car ownership consistently contributed to the wealth score positively in all nations. Due to differences in context and lifestyle across countries, the impact on wealth will vary. It is reasonable to allow for this variability and focus on worrying signs such as when variables that are strong wealth indicators become predictors of poverty from one country to another. The analysis was performed using STATA/SE 18.0 [23].

### Validation of the universal wealth index using a macroeconomic approach

To examine the validity of our PCA-based universal wealth index, we compared it with an alternative regionally standardized wealth index constructed from local wealth indexes and transformed it using a country-specific macroeconomic adjustment factor. As raw local scores could not be directly compared with our universal scores, they needed to be standardized because the ranges of the country's local PC1 scores vary and some are negative. To make them comparable, the local score ranges were converted to a unified scale from 1 to 100 as follows:

- The lowest PC1 score in a country is set to 1.

- The highest PC1 score in a country is set to 100.

- All other values were linearly scaled between 1 and 100.

$$Standardized\ PC1\ Score = \frac{PC1\ Household - min(PC1\ Country)}{max(PC1\ Country) - min(PC1\ Country)} * 100$$

Rescaling addressed score range differences, but they were not yet suitable for pooling these scores to assume a universal score. These countries have notable disparities in national wealth. This study proposed converting country-specific wealth scores to macroeconomic-adjusted scores. This involved adjusting each household's score by the ratio of their local GDP per capita to a reference value, which in this case was Tanzania's GDP per capita. The adjustment factor was then used to convert households' local rankings in each country to regional rankings. The factor was based on 2019 GDP per capita, PPP (current international $) from the WDI database [22], chosen as the median year for the study's data collected between 2016 and 2021. The adjustment factor that accounts for cross-country economic differences was calculated using the formula:

$$Macroeconomic\ Adjuster = \frac{Country\ GDP\ per\ capita,\ PPP(2019)}{Median\ Country\ GDP\ per\ capita,\ PPP(Tanzania,\ 2019)}$$

Tanzania, as the median country in terms of GDP per capita amongst the 15 participating PHIA countries, was assigned the adjustment factor of 1. This transformation of locally standardized wealth scores to a regional score provided a regionally standardized score whereby all households could be separately ranked in a nation agnostic fashion.

$$Globally\ Standardized\ Wealth\ Score = Locally\ Standardized\ PC1\ Score * Macroeconomic\ Adjuster$$

The macroeconomic adjustment step enabled household mobility on a regional scale. It increased the wealth scores of richer countries with GDP per capita above the median using a multiplier greater than 1, while it decreased the scores for countries below the median. This resulted in a regional, country-agnostic wealth score. Households were then ranked based on this doubly transformed wealth score, and the population was classified into five equally sized wealth quintiles (Q1–Q5). This created two separate rankings that were regional in nature: the ranking generated by our own PCA method

that resulted in a universal wealth score based on the pooled samples of 15 PHIA surveys, and the regionally standardized ranking that was provided by proper adjustments to the local raw scores. This study's validation process involved visual and statistical comparisons of household mobility in the two ranking systems.

### Ethics statement

Patient consent for publication: **Not applicable**

Ethics approval: This study used data from the Population-based HIV Impact Assessments (PHIA) survey from ICAP at Columbia University. All PHIA survey protocols, consent forms, screening forms, refusal forms, referral forms, recruitment materials and questionnaires were reviewed and approved by in-country ethics and regulatory bodies and the institutional review boards of Columbia University Medical Center, Westat, and the U.S. Centers for Disease Control and Prevention. Further approval for this study was not required since data are available to all researchers by request.

## Results

Principal component analysis of the 24 household asset and demographic variables produced six components with eigenvalues greater than one. The first component had the largest eigenvalue (5.57) and explained 23.2% of the variance, while the next five components explained between 4.7% and 10.2% each. Consistent with prior applications of PCA-based wealth indices, we used the first component to generate the wealth index, as it best represents the underlying socioeconomic gradient [15,20]. The 24 included variables demonstrated acceptable internal consistency, with a Cronbach's alpha of 0.73. Table 2 presents the eigenvalues and proportion of variance explained for all 24 principal components.

To better understand the distribution and influence of household characteristics on wealth, we evaluated each variable in the PCA. A summary of these features is presented in the study and shows how each contributes to the universal wealth index calculated from the pooled PHIA dataset. Table 3 lists the household characteristics variables along with their rates (or mean or median where appropriate), the range of the variable, the PC1 assigned score to the variable, and the relative importance of the variable where both wealth increasing and wealth decreasing variables are in the ranking. This ranking is from the PCA model applied to pooled PHIA surveys, meaning the wealth score contribution of some items such as housing materials may vary across the 15 countries. The wealth decreasing indicators, which have negative PC1 weights, are ranked from 18 to 24 and are shown in parentheses. Since a household's wealth score is calculated as a linear combination of all variables, households possessing these items receive a lower wealth score. Overall, access to electricity contributed most to the wealth score and ranked 1st in the list. Conversely, number of children under 18 years of age decreased the score most and was ranked 24th, the lowest position.

As indicated, the PC1 weights and rankings in Table 3 are derived from the pooled universal model. Stability of the relative rankings for all 24 household variables was assessed using Spearman's rank correlation [24], a non-parametric approach. We ran 15 pair-wise rank correlation tests between each country's individual PC1 ranking and the ranking generated by PC1 for the pooled model. Table 4 displays the pairwise Spearman's results, suggesting high correlation values all over, meaning that the rankings remain stable. The correlations range from 79% in Eswatini to as high as 97% in Malawi. For example, in Malawi, the PC1 generated importance of household characteristics in predicting wealth, was nearly identical to those proposed by the PC1 in the universal pooled model.

Fig 1 presents the relative importance of household characteristics in predicting wealth for 15 country-specific models and the universal model applied to the pooled sample. In this study, Uganda was the first country where the PCA-based wealth score was implemented, so households' characteristics weights and importance in Uganda were used as the base case. All other countries were depicted based on Uganda's ranking. Characteristics with a negative impact on the wealth score in Uganda are represented by orange and placed in the lower part of the graph. There is no significant number of swings observed for the wealth indicators. Positive-negative sign reversals are infrequent. Roofing material in Rwanda switches sides and becomes a wealth-decreasing indicator. The overall behavior of the wealth indicators demonstrates

**Table 2. Eigenvalues and variance explained by principal components.**

| Component | Eigenvalue | Proportion of Variance | Cumulative Variance |
|---|---|---|---|
| Comp1 | 5.57 | 23.2% | 23.2% |
| Comp2 | 2.46 | 10.2% | 33.4% |
| Comp3 | 1.70 | 7.1% | 40.5% |
| Comp4 | 1.40 | 5.8% | 46.3% |
| Comp5 | 1.19 | 4.9% | 51.3% |
| Comp6 | 1.13 | 4.7% | 55.9% |
| Comp7 | 0.96 | 4.0% | 59.9% |
| Comp8 | 0.89 | 3.7% | 63.7% |
| Comp9 | 0.84 | 3.5% | 67.2% |
| Comp10 | 0.83 | 3.5% | 70.7% |
| Comp11 | 0.74 | 3.1% | 73.8% |
| Comp12 | 0.71 | 2.9% | 76.7% |
| Comp13 | 0.68 | 2.8% | 79.5% |
| Comp14 | 0.65 | 2.7% | 82.2% |
| Comp15 | 0.61 | 2.6% | 84.8% |
| Comp16 | 0.60 | 2.5% | 87.3% |
| Comp17 | 0.58 | 2.4% | 89.7% |
| Comp18 | 0.54 | 2.3% | 91.9% |
| Comp19 | 0.50 | 2.1% | 94.0% |
| Comp20 | 0.47 | 2.0% | 96.0% |
| Comp21 | 0.39 | 1.6% | 97.6% |
| Comp22 | 0.32 | 1.3% | 98.9% |
| Comp23 | 0.14 | 0.6% | 99.6% |
| Comp24 | 0.10 | 0.4% | 100% |

Note: Comp = Component.

a satisfactory level of congruence across all countries and the universal pooled model. This study's stability test results provided sufficient assurance regarding the consistency of the behavior of individual wealth components across numerous countries.

After testing the stability of household characteristics, we further validated our universal ranking system by comparing the mobility of households from local to regional rankings made possible by our PCA approach and separately by the macroeconomic adjusted local scores. We expect household movements driven by macroeconomic adjustments to align with relocations using the same PCA method applied directly to the pooled data without any adjustment. Figs 2–5 show the wealth movements of households from Mozambique and Botswana. Five pairs of bars compare local and regional wealth for five quintiles. Solid left bars represent household rankings within their country, while right bars show their regional quintile positions. One would expect both approaches to shift most lower and lower-middle class households in Botswana to upper-middle and upper wealth quintiles regionally. This is shown in Fig 2, where purple colors, indicating more wealth, dominate the right bars. A significant macroeconomic adjustment factor applied to all households in Botswana turns left bars into light blue and purple bars for almost all households. This movement is shown in Fig 3. Our PCA approach, which operates without information regarding the relative wealth of the nations involved and remains entirely unsupervised, introduces significant amounts of purple and light blue colors to most of the right bars. These elevated levels of purple and light blue in the landing right bars suggest that the universal PCA model is classifying a majority of Botswana households as belonging to the middle-upper and upper class in the regional context as part of its relocation decisions. Additionally as

**Table 3. Households characteristics and assets statistics and their importance in creation of the universal wealth index.**

| Variable Name & Description | Rate*/ Mean/ Median (Universal Pooled Model) | Lowest | Highest | PC1 weight | Factor's Importance Rank** |
|---|---|---|---|---|---|
| havmobl: possession of Tel/ Mobile | 66.90% | Zam*** − 1.5%, | Cot – 91.3% | 0.12 | 16th |
| haverad: possession of radio | 48.60% | Mal – 30.1% | Ken – 63.6% | 0.15 | 13th |
| havetele: possession of television | 33.83% | Mal – 10.5% | Bot – 61.6% | 0.29 | 4th |
| havfrig: possession of refrigerator | 19.24% | Rwa – 4.1% | Bot – 64.1% | 0.26 | 7th |
| haveelect: access to electricity | 45.37% | Mal – 13.4% | Eth – 91% | 0.32 | 1st |
| owncar: ownership of a car | 8.68% | Mal – 2.4% | Bot – 41.8% | 0.16 | 12th |
| ownbike: ownership of bicycle | 21.22% | Les – 4.1% | Tan – 41.1% | −0.08 | (19th) |
| ownmoto: ownership of moto | 8.37% | Les – 0.8% | Cot – 34% | 0.01 | 17th |
| owngtshp: ownership of goats/ sheep | 24.38% | Eth – 8.9% | Zim – 39.6% | −0.13 | (21st) |
| owncows: ownership of cows | 22.42% | Mal – 4.8% | Moz – 66.4% | −0.09 | (20th) |
| ownplry: ownership of poultry | 39.52% | Eth – 10.4% | Zim – 55% | −0.16 | (23rd) |
| watersource | 7 | Bot, Ken, Mal, Uga, Zam – 6 | Cam, Esw, Eth, Nam, Tan – 9 | 0.25 | 8th |
| cookingfuel | 2 | Cam, Cote, Ken, Mal, Nam, Rwa, Uga, Zam, Zim – 1 | Bot, Les – 4 | 0.31 | 3rd |
| toilettype | 2 | Cam, Cot, Esw, Eth, Les, Mal, Rwa, Uga, Zam, Zim – 2 | Tan – 7 | 0.13 | 15th |
| matfloor: material of floor | 3 | Mal, Rwa, Tan, Zam – 1 | Ken, Nam – 5 | 0.29 | 5th |
| matroof: material of roof | 4 | Cot, Est, Moz, Rwa, Tan, Zam – 2 | Cam, Eth – 9 | 0.22 | 9th |
| matexwalls: material of exterior walls | 8 | Eth, Rw – 5 | Nam – 2 | 0.17 | 11th |
| childcount: Count of children aged 0–17 on the roster | 2.05 | Les – 1.17 | Uga – 2.44 | −0.17 | (24th) |
| rostercount: Count of people on household roster | 4.39 | Les – 3.28 | Rwa – 5.09 | −0.16 | (22nd) |
| roompercapita: roomsleep/ rostercount, | 0.61 | Zam – 0.50 | Bot – 0.86 | 0.14 | 14th |
| Urban | 41.4% | Mal – 17.8% | Eth – 100% | 0.31 | 2nd |
| urbanroompercapita: urban*roompercapita | 0.27 | Mal – 0.11 | Bot – 0.53 | 0.28 | 6th |
| householdheadmale | 56.8% | Esw – 41.1% | Cot – 81.1% | −0.03 | (18th) |
| hholdhead_education_years | 7 | Ken – 4 | Zim – 13 | 0.22 | 10th |

* Rankings in parenthesis are factors with negative weights.

** For variables with ordinal ranking, we used the median: (watersource, toilettype, matfloor, matroof, matexwalls, cookingfuel, hholdhead_education_years).

*** Country abbreviations: Bot = Botswana, Cam = Cameroon, Cot = Cote D'Ivoire, Esw = Eswatini, Eth = Ethiopia, Ken = Kenya, Les = Lesotho, Mal = Malawi, Moz = Mozambique, Nam = Namibia, Rwa = Rwanda, Tan = Tanzania, Uga = Uganda, Zam = Zambia, Zim = Zimbabwe.

shown in Fig 4, for Mozambique, we also observe dark blue and orange colors in the right bars for the middle and upper-class households. This movement indicates that those in the upper wealth distribution within poor countries have shifted to the lower end of the wealth distribution on a regional scale. In other words, the upward/downward relocation made possible by our universal PCA-based ranking system for the households coming from high/low income countries shown in Fig 4

**Table 4. Spearman's rank correlation of the household characteristics rankings in PC1 between the universal model and individual country's models.**

| Country | Spearman's Ranked Correlation |
| --- | --- |
| **Botswana** | 0.89 |
| **Cameroon** | 0.91 |
| **Cote D'Ivoire** | 0.86 |
| **Eswatini** | 0.79 |
| **Ethiopia** | 0.81 |
| **Kenya** | 0.85 |
| **Lesotho** | 0.95 |
| **Malawi** | 0.97 |
| **Mozambique** | 0.95 |
| **Namibia** | 0.91 |
| **Rwanda** | 0.86 |
| **Tanzania** | 0.96 |
| **Uganda** | 0.95 |
| **Zambia** | 0.94 |
| **Zimbabwe** | 0.90 |

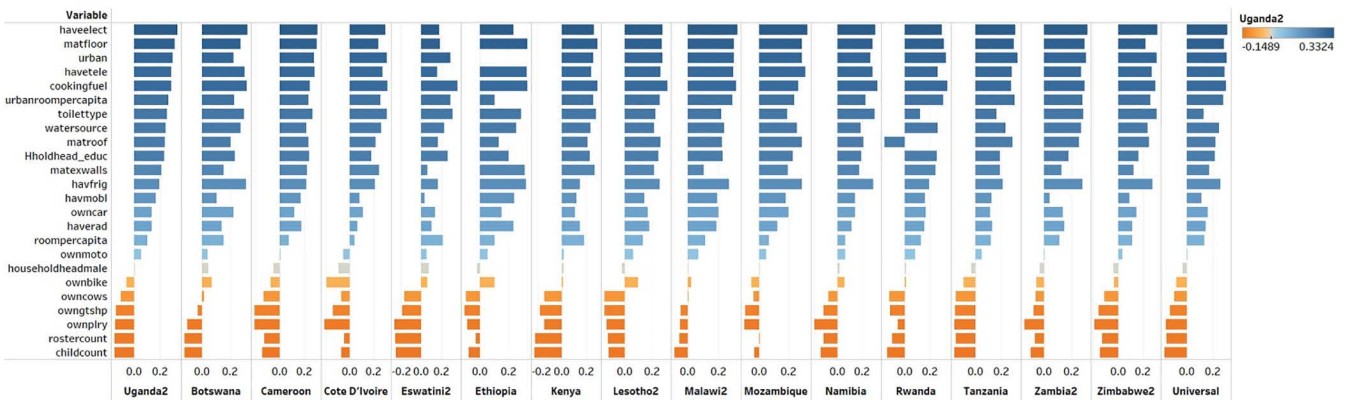

Sum of Uganda2, sum of Botswana, sum of Cameroon, sum of Cote D'Ivoire, sum of Eswatini2, sum of Ethiopia, sum of Kenya, sum of Lesotho2, sum of Malawi2, sum of Mozambique, sum of Namibia, sum of Rwanda, sum of Tanzania, sum of Zambia2, sum of Zimbabwe2 and sum of Universal for each Variable.  Color shows sum of Uganda2.

**Fig 1. Household characteristics as predictors of wealth in country-specific models (bar charts 1 through 15) and the universal model (the rightmost bar chart).** Note: Results were generated based on the PC1 scores of each household wealth proxy variable. Uganda was used as the reference country. PHIA Survey years: Uganda2 (PHIA2 - 2020), Botswana (PHIA1 - 2021), Cameroon (PHIA1 - 2017), Cote D'Ivoire (PHIA1 - 2017), Eswatini2 (PHIA2 - 2020), Ethiopia (PHIA1 - 2017), Kenya (PHIA1 - 2018), Lesotho2 (PHIA2 - 2020), Malawi2 (PHIA2 - 2020), Mozambique (PHIA1 - 2021), Namibia (PHIA1 - 2017), Rwanda (PHIA1 - 2018), Tanzania (PHIA1 - 2016), Zambia2 (PHIA2 - 2021), Zimbabwe2 (PHIA2 - 2020).

is consistent with the movements suggested by the macroeconomic adjustment approach as seen in Fig 5. In summary, the entire distribution of households from lower-income countries tend to relocate to lower universal wealth quintiles while the entire distribution of households originally from higher income countries move upward to occupy the upper middle and upper quintiles in a regional setting, and this dynamic was consistently observed in both PCA direct method over the pooled data and the macroeconomic adjusted approach. Figs 2–5 depict the comparison of the upward and downward mobility of households in their local scale vs their position as they step into a regional stage using two approaches of macroeconomic adjustment vs pooled PCA scores.

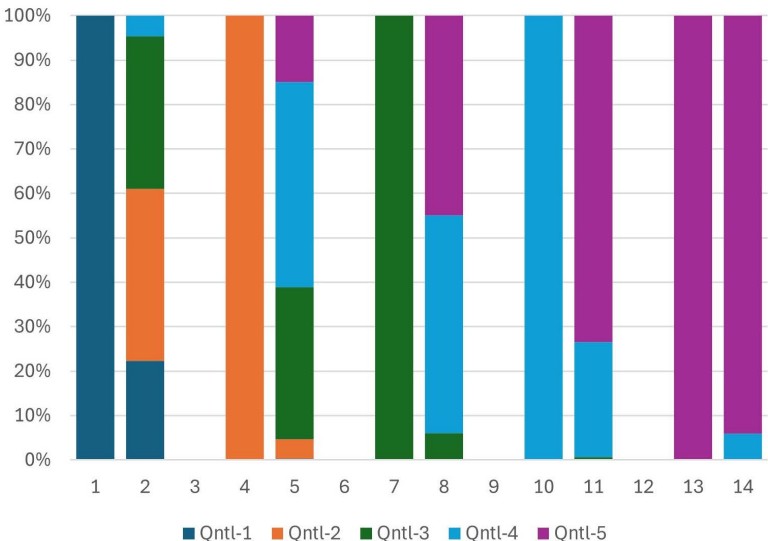

**Fig 2. Botswana- PCA method.**

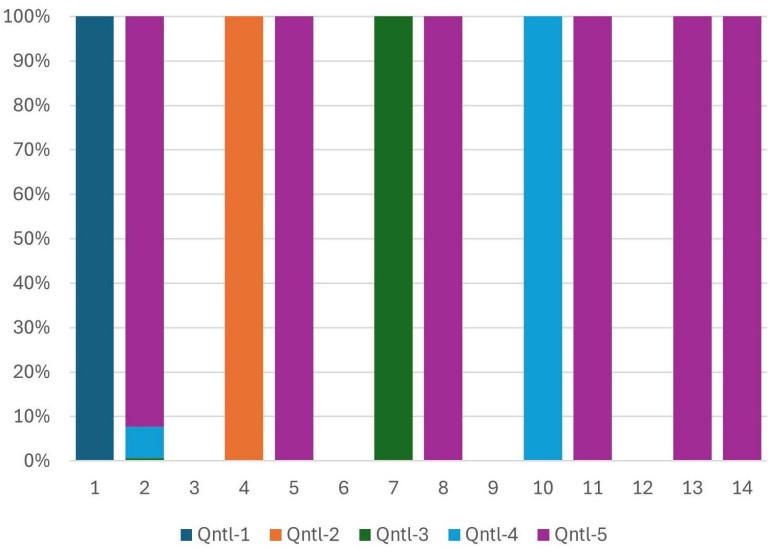

**Fig 3. Botswana- Macroeconomic adjustment method.**

## Discussion

Most policy research that studies the relation between health and wealth focuses on local country data and information. However, as the need for reallocating donor and international funds grows, and transitioning of the HIV and other health program investments to local governments becomes inevitable, the tools and methods for monitoring evaluation research with a global agenda becomes a necessity. This study is an effort to fulfill a part of such a global research agenda. Classic theories of economics of health and healthcare [25] always discuss the inevitable efficiency-equity tradeoff in most of the health policies. The universal eligibility criteria proposed here seem promising in achieving both goals with no tradeoff.

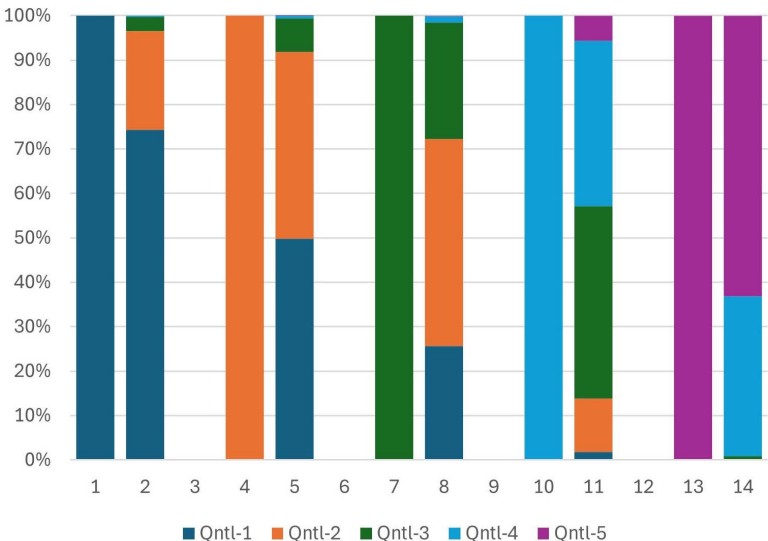

**Fig 4. Mozambique- PCA method.**

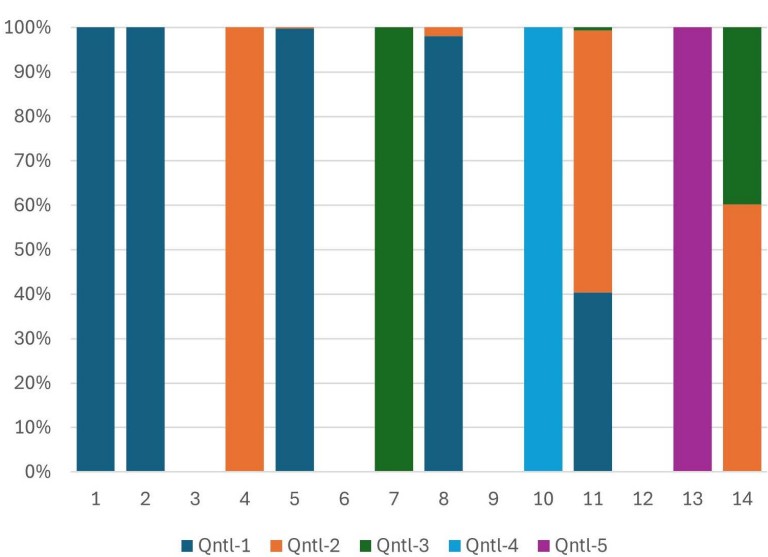

**Fig 5. Mozambique- Macroeconomic adjustment method.**

By using PCA with multiple data sources and demonstrating stable loadings across countries, our approach eliminates the need for macroeconomic adjustment. This cross-country scale maintains consistent relative positioning of household indicators, making it suitable not only for national analysis but also for supporting regional policy reforms and enabling more equitable and efficient resource allocation, particularly for multinational programs like PEPFAR. While we refer to the development of a household wealth index, it is important to clarify that this study focuses on 15 countries within sub-Saharan Africa. The resulting index is therefore most appropriate for use in this regional context. Household assets and their relative contribution to the wealth index may vary substantially across world regions because of cultural, socioeconomic,

geographical and climatic differences; for example, indicators relevant in sub-Saharan Africa may have limited comparability in Asia or Latin America. As such, the framework we propose should be understood as a methodological foundation that can be adapted to construct regional indices tailored to other contexts. To examine the applicability of the approach, we used our universal wealth index to evaluate health outcomes across different countries and guide resource allocation. To account for differences in country population size when applying the pooled wealth index to community health worker (CHW) data, we adjusted sample representation using a two-step procedure. First, we calculated the PHIA survey sample rate relative to each country's population and identified the median rate across the 15 countries. We then resampled each country's data so that its sample size corresponded to the median rate, using bootstrapping to oversample countries with smaller relative sample sizes and undersample those with larger relative sample sizes. To maintain socioeconomic balance, resampling was stratified by wealth quintile to ensure that each quintile represented 20% of the resampled dataset. From this balanced data-set we generated PCA-based wealth scores and ranked households into five universal wealth quintiles.

Assuming assistance eligibility for the poorest quintile, the shift from local to universal ranking increased representation of poorer countries in the lowest quintile, making previously ineligible households eligible for support. Conversely, fewer households from richer countries were classified as the poorest. Using PEPFAR 2024 data (see Table in S1 Table) for Community Health Workers (CHWs), excluding Lesotho, Rwanda, and Ethiopia, we found that poorer countries like Mozambique, Malawi, and Uganda received more funding, while richer countries like Eswatini, Namibia, and Botswana saw reductions. This reallocation was pro-poor, favoring the bottom third of the wealth index. The total expenditure for CHWs under the local schema was $58.2 million, while the universal schema cost $50.5 million—a net saving of $7.7 million or 15%. These savings are due to reallocating funds more efficiently, with lower labor costs in poorer countries, demonstrating that a universal framework leads to smarter, fairer, and more efficient resource allocation. Country-specific indices are useful tools for national program planning, allocating domestic resources and addressing within-country inequities, whereas regional indices are more suitable for cross-country comparisons and donor prioritization. It is also important to acknowledge ongoing efforts to make wealth measurement more efficient, such as the EquityTool, which provides simplified, validated proxies for rapid assessment of wealth while maintaining comparability [26]. Several efforts have been made to develop cross-country wealth indices, including PCA-based methods [15], approaches combining household relative wealth with national wealth distributions [27], anchoring techniques [28], and more recent random forest machine-learning models leveraging remote sensing data [29], most of which rely on DHS surveys. These initiatives have advanced the comparability of household wealth across countries in both relative and absolute terms. Our approach aligns methodologically with these efforts but differs in its data source, as it is calibrated to PHIA data from 15 sub-Saharan African countries, enabling inter-country comparisons that are directly relevant to HIV programmatic needs. As demonstrated earlier, creation of a universal ranking schema is readily available using macroeconomic adjusters created using countries' relative PPP-based per capita income. A legitimate question is, what is the real merit of a PCA based ranking system with so much methodological complexity. We believe the PCA method is more appropriate because in global health research the unit of analysis shifts away from classic sectors of a macro economy towards micro units and agents such as households and individuals. Almost all cross-country macroeconomic adjusters apply a single multiplier factor to the entire population of a country with no regard to important microeconomics variables. These adjusters function in a pure mechanical way leading to a blind and bulk upward/downward movement of the entire population as the country enters into a regional stage. Such indiscriminate movement is based on the assumption that wealth and human capital inequalities are not only negligible but also almost identical across countries. However, GDP by definition is the sum of the value added generated by all economic sectors. Imagine two economies, A and B where A's GDP is 10% higher. A's income comes mostly from petrodollars, controlled by the governments and closely affiliated elite firms. B's GDP comes from a highly participatory economy with strong household production and a strong and diverse labor force participation by the general population. A macroeconomic adjuster sees the economic wellbeing of an average citizen in A being always better than an average citizen of B simply because their PPP-based GDP per capita is better. A PCA-based ranking system however does a far

superior job in appropriately classifying the marginalized households in the oil rich country with a standard of living far worse than their average citizen in the universally poor quintile, and probably a fewer percentage of the B's households within the same quintile. In other words, the PCA approach in a regional stage has an implicit ability in taking account of inequalities in households' wealth and human capital which gives the method a superior specificity in targeting the neediest households with no regard to geographical boundaries nor their rather blind GDP per capita measures.

We assessed the comparability of our PCA-based wealth quintiles with the original PHIA quintiles prior to pooling. On a country-by-country basis, the correlations ranged from 0.78 in Eswatini to 0.94 in Lesotho, indicating strong within-country consistency. After pooling the data, the correlation between the pooled PCA regional quintiles and the original PHIA quintiles declined to 0.66. This lower correlation is anticipated, as the regional index reflects households' relative socioeconomic positions across multiple countries. Consequently, households from wealthier countries tended to shift upward in the regional distribution, whereas households from poorer countries tended to shift downward. This re-ranking highlights the ability of the regional index to capture wealth differentials across national boundaries. Importantly, similar upward and downward mobility patterns were observed when applying exogenous macroeconomic scalars, such as GDP per capita, further validating the regional index's capacity to account for cross-country disparities.

Furthermore, while this technical proof of concept was implemented using PHIA surveys, it can be similarly tested and validated on other comparable datasets and surveys, particularly in contexts requiring pooled data from multiple nations.

## Supporting information

**S1 Table. Movement of households from local to universal quintile with its associated tracking of funding reallocation and cost saving.**
(DOCX)

## Acknowledgments

The authors thank Stephen McCracken, Drew Voetsch, Faith L. Ussery, Paul Stupp, and Abraham D. Ater of the U.S. Centers for Disease Control and Prevention and VS Senthil Kumar of Brandeis University for their invaluable input throughout the methodological design of this paper. The authors also thank Clare L. Hurley for her editorial assistance.

## Author contributions

**Data curation:** Moaven Razavi, Collins Gaba.

**Formal analysis:** Moaven Razavi, William Crown, Allyala Nandakumar.

**Funding acquisition:** Allyala Nandakumar.

**Methodology:** Moaven Razavi, Collins Gaba, William Crown, Allyala Nandakumar.

**Supervision:** Moaven Razavi, Allyala Nandakumar.

**Validation:** Moaven Razavi, William Crown, Allyala Nandakumar.

**Visualization:** Moaven Razavi, William Crown, Allyala Nandakumar.

**Writing – original draft:** Moaven Razavi, Collins Gaba, William Crown, Allyala Nandakumar.

**Writing – review & editing:** Moaven Razavi, Collins Gaba, William Crown, Allyala Nandakumar.

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
