## [Decision Letter · Decision Letter 0]

2 Sep 2025

Dear Dr. Razavi,

Thank you for submitting your manuscript to PLOS ONE. After careful consideration, we feel that it has merit but does not fully meet PLOS ONE’s publication criteria as it currently stands. Therefore, we invite you to submit a revised version of the manuscript that addresses the points raised during the review process.

**Two experts have reviewed the manuscript noting limitations. In particular, in terms of scope, it seems that the authors are overselling their attempt, which is not a methodological advance since similar indices have been calculated with similar methodology elsewhere and this previous research should be cited in the current text, locating the specific novelties. In my opinion the lie mainly in the data set and I see it more as a data-set specific contribution. Note also that "global" in the title is not accurate. This is a regional index for Sub-Saharan Africa.**

We look forward to receiving your revised manuscript.

Kind regards,

José Antonio Ortega, Ph.D.

Academic Editor

PLOS ONE

**Journal Requirements:**

1. When submitting your revision, we need you to address these additional requirements. Please ensure that your manuscript meets PLOS ONE's style requirements, including those for file naming. The PLOS ONE style templates can be found at https://journals.plos.org/plosone/s/file?id=wjVg/PLOSOne_formatting_sample_main_body.pdf and https://journals.plos.org/plosone/s/file?id=ba62/PLOSOne_formatting_sample_title_authors_affiliations.pdf 2. Please note that PLOS One has specific guidelines on code sharing for submissions in which author-generated code underpins the findings in the manuscript. In these cases, we expect all author-generated code to be made available without restrictions upon publication of the work. Please review our guidelines at https://journals.plos.org/plosone/s/materials-and-software-sharing#loc-sharing-code and ensure that your code is shared in a way that follows best practice and facilitates reproducibility and reuse. 3. Thank you for stating the following financial disclosure: The contents of this manuscript were developed under grant INV-046280 from the Bill & Melinda Gates Foundations. The content is solely the responsibility of the authors and does not necessarily represent the official views of the funding foundation.   Please state what role the funders took in the study.  If the funders had no role, please state: "The funders had no role in study design, data collection and analysis, decision to publish, or preparation of the manuscript." If this statement is not correct you must amend it as needed. Please include this amended Role of Funder statement in your cover letter; we will change the online submission form on your behalf. 4. When completing the data availability statement of the submission form, you indicated that you will make your data available on acceptance. We strongly recommend all authors decide on a data sharing plan before acceptance, as the process can be lengthy and hold up publication timelines. Please note that, though access restrictions are acceptable now, your entire data will need to be made freely accessible if your manuscript is accepted for publication. This policy applies to all data except where public deposition would breach compliance with the protocol approved by your research ethics board. If you are unable to adhere to our open data policy, please kindly revise your statement to explain your reasoning and we will seek the editor's input on an exemption. Please be assured that, once you have provided your new statement, the assessment of your exemption will not hold up the peer review process. 5. Please include captions for your Supporting Information files at the end of your manuscript, and update any in-text citations to match accordingly. Please see our Supporting Information guidelines for more information: http://journals.plos.org/plosone/s/supporting-information. 6. If the reviewer comments include a recommendation to cite specific previously published works, please review and evaluate these publications to determine whether they are relevant and should be cited. There is no requirement to cite these works unless the editor has indicated otherwise. 

Reviewers' comments:

**Comments to the Author**

1. Is the manuscript technically sound, and do the data support the conclusions?

Reviewer #1: Yes

Reviewer #2: Yes

2. Has the statistical analysis been performed appropriately and rigorously?

Reviewer #1: Yes

Reviewer #2: Yes

3. Have the authors made all data underlying the findings in their manuscript fully available?

Reviewer #1: No

Reviewer #2: Yes

4. Is the manuscript presented in an intelligible fashion and written in standard English?

Reviewer #1: Yes

Reviewer #2: Yes

**Reviewer #1: ** This is an interesting approach to examining regional wealth disparities in Africa. There are several areas where I feel that the submission could be improved.

1. Title and text - The authors refer to the "Global Household Wealth Index" but what they have created is a regional index for sub-Saharan Africa, using 15 SSA countries. While I am sure there is some variability between these countries on the relative value of the different assets, if one was to apply this to other regions, there is likely to be much more variability. The authors should emphasize the appropriateness of applying this particular index to the SSA region and that further regional indices should be developed if the methodology was to be used in other regions (e.g. Asia, Latin America).

2. Background - The background would benefit from some discussion about the different approaches to assessing wealth and income, with both relative measures like the wealth quintiles and absolute measures like the World Bank poverty thresholds. There are pros and cons to using relative vs. absolute measures and this methodology is essentially trying to create a regional relative index so that the countries who have a higher absolute level of poverty (greater percentage under the poverty line) are better represented when thinking about allocating development resources. You touch on this in the discussion section, but it would be good to set up the overall concepts in the background.

3. Methods - How were the survey weights for the individual surveys were handled when combining to a larger dataset. Was a population-size weight applied to adjust for disproportionate sample sizes relative to the actual country populations?

4. Results - When running the combined PCA, how many factors did the data load into? Did only PC1 have an eigenvalue above 1? It would be preferable if you shared that output. Also, it's clear you started with 24 variables - were all included in the index? Please specify and provide the Cronbach's alpha for the included variables.

5. Discussion - It would be useful to include some discussion about when it would be appropriate to use a regional vs. country level wealth index for policy purposes. It might also be good to acknowledge somewhere all the work that has been done to make capturing wealth more efficient, like the EquityTool (https://equitytool.org/).

**Reviewer #2: ** The paper describes the development of an asset-based wealth index using PCA that applies across countries. Many other asset-based wealth indices have already been developed and tested that accomplish the same goal (see citations below for just a few). More recently, there have been a number of machine-learning approaches that have been used to create asset-based wealth indices that are comparable across countries (Wang and Li 2024 for just one example).

It’s not clear from the article how the current paper improves upon those existing measures. At a minimum, the paper should examine empirically how the new measure performs better than existing approaches. Given that it uses similar methods to existing measures, I’m not sure how it will make a major improvement, but an empirical demonstration would support the value added of the approach outlined here.

Smits, J., & Steendijk, R. (2015). The international wealth index (IWI). Social indicators research, 122(1), 65-85.

Hruschka, D. J., Gerkey, D., & Hadley, C. (2015). Estimating the absolute wealth of households. Bulletin of the World Health Organization, 93, 483-490.

Rutstein, S. O., & Staveteig, S. (2014). Making the demographic and health surveys wealth index comparable (Vol. 9). Rockville, MD: ICF international.

Wang, M., & Li, X. (2024). Estimation of long time-series fine-grained asset wealth in Africa using publicly available remote sensing imagery. International Journal of Applied Earth Observation and Geoinformation, 135, 104269.

**Do you want your identity to be public for this peer review?** For information about this choice, including consent withdrawal, please see our Privacy Policy

Reviewer #1: No

Reviewer #2: No

---

## [Author Response · Author response to Decision Letter 1]

23 Sep 2025

We have uploaded the Response to Reviewers in the upload section and we are copying the same responses here:

Dear Dr. Ortega,

We sincerely thank you and the reviewers for the careful evaluation of our manuscript, “Formulation and Validation of a Regional Household Wealth Index for Sub-Saharan Africa” previously titled “Formulation and Validation of a Global Household Wealth Index.” The feedback provided was highly constructive and has helped us strengthen the manuscript. Below, we provide a point-by-point response to each comment. Please do not hesitate to contact us if further clarification is needed.

Warm regards,

Moaven Razavi, PhD

Research Scientist

The Heller School for Social Policy and Management

Brandeis University

Reviewer #1: This is an interesting approach to examining regional wealth disparities in Africa. There are several areas where I feel that the submission could be improved.

1. Title and text - The authors refer to the "Global Household Wealth Index" but what they have created is a regional index for sub-Saharan Africa, using 15 SSA countries. While I am sure there is some variability between these countries on the relative value of the different assets, if one was to apply this to other regions, there is likely to be much more variability. The authors should emphasize the appropriateness of applying this particular index to the SSA region and that further regional indices should be developed if the methodology was to be used in other regions (e.g. Asia, Latin America).

We thank the reviewer for this important observation. We agree that while our study presents the formulation and validation of a household wealth index based on 15 countries in sub-Saharan Africa, it is most appropriate to interpret this as a regional index rather than a truly global index.

In light of this, we have made the following revisions:

a. Title Revision – We will revise the title to, “Formulation and Validation of a Regional Household Wealth Index for Sub-Saharan Africa” to more accurately reflect the scope of our work.

b. We also changed the use of the term “global/globally” to “regional/regionally” to better reflect the nature of the wealth index.

c. We have also highlighted that while the methodology is generalizable, additional regional indices should be developed if the approach is to be extended to other regions, given differences in socioeconomic and cultural contexts. We made this revision in the discussion section. Revised text now reads, “While we refer to the development of a household wealth index, it is important to clarify that this study focuses on 15 countries within sub-Saharan Africa. The resulting index is therefore most appropriate for use in this regional context. Household assets and their relative contribution to the wealth index may vary substantially across world regions because of cultural, socioeconomic, geographical and climatic differences; for example, indicators relevant in sub-Saharan Africa may have limited comparability in Asia or Latin America. As such, the framework we propose should be understood as a methodological foundation that can be adapted to construct regional indices tailored to other contexts.”

2. Background - The background would benefit from some discussion about the different approaches to assessing wealth and income, with both relative measures like the wealth quintiles and absolute measures like the World Bank poverty thresholds. There are pros and cons to using relative vs. absolute measures and this methodology is essentially trying to create a regional relative index so that the countries who have a higher absolute level of poverty (greater percentage under the poverty line) are better represented when thinking about allocating development resources. You touch on this in the discussion section, but it would be good to set up the overall concepts in the background.

We thank the reviewer for this contribution, and we agree that this will strengthen our background section. We have therefore added the following to the background “Different approaches exist to measure wealth and poverty, including relative measures such as the DHS wealth quintiles [8,9] and absolute measures such as the World Bank’s international poverty thresholds [10,11]. Relative measures are useful for assessing inequality within countries [12], while absolute measures enable cross-country comparisons but may overlook within-country disparities [13]. Our study builds on the relative approach by creating a regional index that allows households across sub-Saharan Africa to be ranked on the same scale.”

3. Methods - How were the survey weights for the individual surveys handled when combining to a larger dataset. Was a population-size weight applied to adjust for disproportionate sample sizes relative to the actual country populations?

We thank the reviewer for raising this point. We acknowledge that not weighting by population may appear to overrepresent smaller countries or underrepresent larger ones. However, previous work constructing an International Wealth Index from 97 DHS surveys demonstrated that the results are highly robust to this choice. Smits and Steendijk (2015) tested both extremes—population-size weighting versus equal country weighting—and found that the resulting indices were nearly identical (Pearson correlation 0.999). Suggesting that PCA-based wealth indices are very stable across alternative weighting schemes.

In line with that evidence, we proceeded with equal weighting across PHIA countries. We have clarified this rationale in the Methods section and noted the limitation that unequal country population sizes were not explicitly accounted for. We therefore added this statement to the methods section “Data from all countries were pooled without applying population-size weights, following approaches used in prior cross-country wealth index research [15], which has shown that PCA-based wealth indices are highly robust to whether countries are weighted equally or by population size.”

Additionally, when applying the pooled wealth index to community health worker (CHW) data across the 15 PHIA countries in our discussion section, we accounted for differences in national population size to avoid disproportionate representation of smaller or larger countries. We will add this methodological approach to the discussion section with the statement, “To account for differences in country population size when applying the pooled wealth index to community health worker (CHW) data, we adjusted sample representation using a two-step procedure. First, we calculated the PHIA survey sample rate relative to each country’s population and identified the median rate across the 15 countries. We then resampled each country’s data so that its sample size corresponded to the median rate, using bootstrapping to oversample countries with smaller relative sample sizes and undersample those with larger relative sample sizes. To maintain socioeconomic balance, resampling was stratified by wealth quintile to ensure that each quintile represented 20% of the resampled dataset.”

4. Results - When running the combined PCA, how many factors did the data load into? Did only PC1 have an eigenvalue above 1? It would be preferable if you shared that output. Also, it's clear you started with 24 variables - were all included in the index? Please specify and provide the Cronbach's alpha for the included variables.

This is an excellent point and we appreciate you touching on it, we believe this will strengthen our results section. We have therefore added the following statement to the results section as well as the Eigenvalue output. “Principal component analysis of the 24 household asset and demographic variables produced six components with eigenvalues greater than one. The first component had the largest eigenvalue (5.57) and explained 23.2% of the variance, while the next five components explained between 4.7% and 10.2% each. Consistent with prior applications of PCA-based wealth indices, we used the first component to generate the wealth index, as it best represents the underlying socioeconomic gradient.[9,14] The 24 included variables demonstrated acceptable internal consistency, with a Cronbach’s alpha of 0.73.”

5. Discussion - It would be useful to include some discussion about when it would be appropriate to use a regional vs. country level wealth index for policy purposes. It might also be good to acknowledge somewhere all the work that has been done to make capturing wealth more efficient, like the EquityTool (https://equitytool.org/ [equitytool.org]).

We thank the reviewer for this suggestion. We have added this paragraph to the Discussion section. “Country-specific indices are useful tools for national program planning, allocating domestic resources and addressing within-country inequities, whereas regional indices are more suitable for cross-country comparisons and donor prioritization. It is also important to acknowledge ongoing efforts to make wealth measurement more efficient, such as the EquityTool,[26] which provides simplified, validated proxies for rapid assessment of wealth while maintaining comparability.

Reviewer #2: The paper describes the development of an asset-based wealth index using PCA that applies across countries. Many other asset-based wealth indices have already been developed and tested that accomplish the same goal (see citations below for just a few). More recently, there have been a number of machine-learning approaches that have been used to create asset-based wealth indices that are comparable across countries (Wang and Li 2024 for just one example).

It’s not clear from the article how the current paper improves upon those existing measures. At a minimum, the paper should examine empirically how the new measure performs better than existing approaches. Given that it uses similar methods to existing measures, I’m not sure how it will make a major improvement, but an empirical demonstration would support the value added of the approach outlined here.

We thank the reviewer for highlighting the important work on cross-country asset-based wealth indices, including the International Wealth Index (IWI) [Smits & Steendijk 2015], the approach by Hruschka et al. [2015], the DHS wealth comparability framework [Rutstein & Staveteig 2014], and recent machine learning methods [Wang & Li 2024]. We have now acknowledged this in our discussion section and how our method contributes to this existing body of work with the following statement. “Several efforts have been made to develop cross-country wealth indices, including PCA-based methods,[15] approaches combining household relative wealth with national wealth distributions, [27]anchoring techniques, [28] and more recent random forest machine-learning models leveraging remote sensing data,[29]most of which rely on DHS surveys. These initiatives have advanced the comparability of household wealth across countries in both relative and absolute terms. Our approach aligns methodologically with these efforts but differs in its data source, as it is calibrated to PHIA data from 15 sub-Saharan African countries, enabling inter-country comparisons that are directly relevant to HIV programmatic needs.”

In regard to empirically demonstrating its comparability to other indices we addressed this by adding this text to the discussion section. “We assessed the comparability of our PCA-based wealth quintiles with the original PHIA quintiles prior to pooling. On a country-by-country basis, the correlations ranged from 0.78 in Eswatini to 0.94 in Lesotho, indicating strong within-country consistency. After pooling the data, the correlation between the pooled PCA regional quintiles and the original PHIA quintiles declined to 0.66. This lower correlation is anticipated, as the regional index reflects households’ relative socioeconomic positions across multiple countries. Consequently, households from wealthier countries tended to shift upward in the regional distribution, whereas households from poorer countries tended to shift downward. This re-ranking highlights the ability of the regional index to capture wealth differentials across national boundaries. Importantly, similar upward and downward mobility patterns were observed when applying exogenous macroeconomic scalars, such as GDP per capita, further validating the regional index’s capacity to account for cross-country disparities.”

We greatly appreciate the time and effort you and the reviewers dedicated to evaluating our manuscript. We believe the revisions and clarifications provided have addressed the concerns raised and have strengthened the contribution of our work. We respectfully resubmit the revised manuscript for your consideration and look forward to your feedback.

---

## [Decision Letter · Decision Letter 1]

14 Oct 2025

Formulation and Validation of a Regional Household Wealth Index for Sub-Saharan Africa

PONE-D-25-30625R1

Dear Dr. Razavi,

We’re pleased to inform you that your manuscript has been judged scientifically suitable for publication and will be formally accepted for publication once it meets all outstanding technical requirements.

Kind regards,

José Antonio Ortega, Ph.D.

Academic Editor

PLOS ONE

Additional Editor Comments (optional):

Both reviewers are satisfied with the revision, so is the academic editor.

Reviewers' comments:

Reviewer's Responses to Questions

**Comments to the Author**

Reviewer #1: All comments have been addressed

Reviewer #2: All comments have been addressed

2. Is the manuscript technically sound, and do the data support the conclusions?

Reviewer #1: Yes

Reviewer #2: Yes

3. Has the statistical analysis been performed appropriately and rigorously?

Reviewer #1: Yes

Reviewer #2: Yes

4. Have the authors made all data underlying the findings in their manuscript fully available?

Reviewer #1: Yes

Reviewer #2: Yes

5. Is the manuscript presented in an intelligible fashion and written in standard English?

Reviewer #1: Yes

Reviewer #2: Yes

Reviewer #1: I am satisfied with the revisions. The authors have adjusted the title, background, and methodology sections accordingly.

Reviewer #2: The authors have addressed all my questions and concerns from an earlier review. It would be very interesting in a future study to see how this measure matches with existing measures developed from other data sources.

**Do you want your identity to be public for this peer review?** For information about this choice, including consent withdrawal, please see our Privacy Policy

Reviewer #1: No

Reviewer #2: No

---

## [Editor Report · Acceptance letter]

PONE-D-25-30625R1

PLOS ONE

Dear Dr. Razavi,

I'm pleased to inform you that your manuscript has been deemed suitable for publication in PLOS ONE. Congratulations! Your manuscript is now being handed over to our production team.

Kind regards,

on behalf of

Dr. José Antonio Ortega

Academic Editor

PLOS ONE